# Enhanced WGAN Model for Diagnosing Laryngeal Carcinoma

**DOI:** 10.3390/cancers16203482

**Published:** 2024-10-14

**Authors:** Sungjin Kim, Yongjun Chang, Sungjun An, Deokseok Kim, Jaegu Cho, Kyungho Oh, Seungkuk Baek, Bo K. Choi

**Affiliations:** 1Department of Artificial Intelligence, Cheju Halla University, Jeju 63092, Republic of Korea; r3dzon3@chu.ac.kr (S.K.); yongnjin@chu.ac.kr (Y.C.); sjan2020@chu.ac.kr (S.A.); 2Research Lab, MTEG, Seoul 03920, Republic of Korea; laurent@mteg.co.kr; 3Department of Otolaryngology-Head and Neck Surgery, Korea University College of Medicine, Seoul 02841, Republic of Korea; jgcho@korea.ac.kr (J.C.); ohkyungho@korea.ac.kr (K.O.); mdbsk@korea.ac.kr (S.B.)

**Keywords:** image segmentation, laryngeal cancer, laryngoscope video, U-Net, WGAN

## Abstract

**Simple Summary:**

This study aimed to enhance the accuracy of detecting laryngeal carcinoma using a modified AI model based on U-Net. The model was designed to automatically identify lesions in endoscopic images. Researchers addressed issues such as mode collapse and gradient explosion to ensure stable performance, achieving 99% accuracy in detecting malignancies. The study found that malignant tumors were detected more reliably than benign ones. This technology could help reduce human error in diagnoses, allowing for earlier detection and treatment. Furthermore, it has the potential to be applied in other medical fields, benefiting overall healthcare.

**Abstract:**

This study modifies the U-Net architecture for pixel-based segmentation to automatically classify lesions in laryngeal endoscopic images. The advanced U-Net incorporates five-level encoders and decoders, with an autoencoder layer to derive latent vectors representing the image characteristics. To enhance performance, a WGAN was implemented to address common issues such as mode collapse and gradient explosion found in traditional GANs. The dataset consisted of 8171 images labeled with polygons in seven colors. Evaluation metrics, including the F1 score and intersection over union, revealed that benign tumors were detected with lower accuracy compared to other lesions, while cancers achieved notably high accuracy. The model demonstrated an overall accuracy rate of 99%. This enhanced U-Net model shows strong potential in improving cancer detection, reducing diagnostic errors, and enhancing early diagnosis in medical applications.

## 1. Introduction

The larynx is essential for speech, breathing, and safeguarding the airway during swallowing. However, laryngopharyngeal cancer (LPC), including the laryngeal and hypopharyngeal cancers, is the second most prevalent malignancy in the head and neck region, responsible for over 130,000 deaths in 2020 [1]. Due to its poor prognosis, early detection and diagnosis are vital for effective treatment through diagnostic tools such as endoscopy. However, diagnosis is difficult for physicians until enough training and experience are accumulated. Besides, human visual observation of laryngeal lesions varies, resulting in a demand for an accurate automatic diagnosis tool for laryngeal endoscopy based on artificial intelligence (AI), which is reproducible and repeatable [2]. The use of image segmentation for lesions of interest in laryngeal endoscopy can especially enhance tumor detection accuracy, potentially enabling earlier diagnosis and improving patient outcomes.

Another major challenge in the diagnosis is open-set classification, which deals with unknown classes. Previous classification studies primarily focused on closed sets to enhance the accuracy for known classes. For example, closed-set face recognition involves learning from a defined group, while open-set recognition must account for unknown classes, complicating the learning process [3,4,5]. This challenge extends to diagnosing cancer from the image data obtained through medical imaging devices.

The imaging devices used in medical facilities include X-ray machines, computed tomography (CT) scanners, magnetic resonance imaging (MRI) machines, and ultrasound devices. These medical imaging tools are essential for diagnosing cancer. Recent imaging technology advancements have led to faster scan times and improved image resolution, resulting in active research on cancer diagnosis utilizing artificial intelligence [6,7].

Recent attempts to utilize medical imaging data, mainly through artificial intelligence, include classifying lesions [8]. Deep learning-based medical image analysis techniques are primarily employed to detect and segment patient lesions [9,10]. For diagnosis, object segmentation is necessary, which refers to dividing the object areas we want to find from the backgrounds in the images or videos. While our eyes can distinguish between the background and object areas, AI technology requires the ability to differentiate subtle variations in brightness. Addressing these differences necessitates training on various types of image data. Furthermore, implementing a deep learning model that can effectively discern these subtle distinctions is crucial.

Among the methods for object detection using neural network technology, semantic image segmentation [11] is the most widely used. Various machine learning algorithms are available for image segmentation, along with numerous open-source tools. Here, the U-Net is utilized as a pixel-level image segmentation model known to be suitable for image classification [12,13].

Laryngeal cancer is a severe and potentially life-threatening disease that affects the larynx, also known as the voice box. The early diagnosis and treatment of this cancer are crucial for improving the patient outcomes and survival rates [14,15]. Modern technology, particularly AI models, has played a significant role in advancing the diagnosis and treatment of laryngeal cancer. The diagnosis of laryngeal cancer primarily relied on clinical examination methods such as indirect laryngoscopy and direct laryngoscopy. These procedures involved using a mirror or a laryngoscope to inspect the larynx for abnormal growths or lesions visually. Additionally, biopsy, in which a sample of suspicious tissue is collected for histopathological examination, has long been a gold standard for confirming the presence of malignancy. Imaging techniques like X-rays, CT, and MRI supported the diagnostic process by providing detailed views of the tumor’s location and extent.

Today, with the advancements in medical technology, the diagnostic methods for laryngeal cancer have significantly improved. High-definition video laryngoscopy and stroboscopy provide more explicit images of the vocal cords, enabling better visualization of potential cancerous growths. In addition to these enhanced visual techniques, AI-based tools and machine learning models have begun to play a vital role in early detection [16,17]. These models can analyze large datasets from imaging scans, detecting subtle patterns and anomalies that might be missed by the human eye.

The integration of these modern technologies has not only improved the accuracy of diagnosis but also enabled earlier intervention, which is critical for better treatment outcomes. The following section will explore the recent technological advancements in laryngeal cancer diagnosis and introduce the relevant research papers.

## 2. Related Work

This chapter describes the various artificial intelligence techniques for laryngeal cancer diagnosis and the WGAN model, which is the basis of our proposed model.

### 2.1. AI Technology for Laryngeal Cancer Diagnosis

AI has shown great promise in improving the accuracy and efficiency of laryngeal cancer diagnosis [17,18]. With the help of machine learning algorithms and deep learning models, AI can analyze large amounts of medical data and assist healthcare professionals in making more accurate diagnoses. In this paper, we will explore some of the latest advancements in AI technology for laryngeal cancer diagnosis and introduce the related research papers.

Computer-aided diagnosis is a technique that uses machine learning algorithms to analyze medical images. This technique can help improve the accuracy of laryngeal cancer diagnosis. Azam et al. created a YOLO-based CNN diagnosis model for laryngeal squamous cell carcinoma using a dataset of 624 video frames from video laryngoscopes [19]. The study achieved a precision of 66% in identifying laryngeal cancer.

Automated laryngeal cancer detection is a technique that uses machine learning algorithms to analyze images of the larynx and identify signs of cancer. This technique can help healthcare professionals make faster and more accurate diagnoses, leading to improved patient outcomes. A study by Esmaeili et al. (2021) introduced a deep learning-based automated laryngeal cancer detection model [20]. The study used a dataset of 8181 contact endoscopy and narrow-band imaging (CE-NBI) images and achieved an accuracy of 83.5% in identifying laryngeal cancer. The researchers concluded that the automated detection model could be valuable in helping healthcare professionals diagnose laryngeal cancer.

Voice analysis uses machine learning algorithms to assess the sound and tone of a patient’s voice, helping identify changes in voice quality that may indicate laryngeal cancer. Sahoo et al. implemented a deep learning-based Mask R-CNN model to identify laryngeal cancer [21]. The study used a dataset of 242 samples and achieved a precision of 98.99% and an F1 score of 97.99% in identifying laryngeal cancer. The researchers concluded that image analysis could be helpful in screening for laryngeal cancer. Like the studies mentioned above, this reflects the ongoing advancements in AI technology, which continuously enhance the accuracy of laryngeal cancer diagnoses.

Automated laryngeal cancer detection, voice analysis, and computer-aided diagnosis are just a few examples of the latest AI techniques available for laryngeal cancer diagnosis. These techniques have been evaluated in various studies, providing valuable insights into their diagnostic capabilities. As AI technology continues to advance, we can expect further improvements in the diagnosis and treatment of laryngeal cancer. In this approach, we propose an improved model specialized for image segmentation and its accuracy in Section 3.

### 2.2. WGAN

The U-Net [22] is a convolutional neural network (CNN) [23] architecture designed for medical image segmentation. It consists of an encoder that downsamples the image and extracts its features, followed by a decoder that upsamples the feature maps and generates a segmentation mask. The U-Net architecture is characterized by a skip connection between the encoder and the decoder, which allows the model to preserve high-resolution features and improve the segmentation accuracy. The Wasserstein GAN (WGAN) is a variant of the GAN architecture proposed by Arjovsky [24]. The WGAN [24,25,26] is a type of generative adversarial network (GAN) [27] that uses the Wasserstein distance to measure the distance between real and fake (generated data) distributions, which provides a more stable and meaningful training signal than that of the traditional GAN loss function. The WGAN architecture has been used in various image generation tasks, such as image synthesis [28] and super-resolution [29]. The WGAN architecture has also been applied to medical image analysis tasks, such as brain tumor segmentation [30] and lung nodule detection [31]. This provides a more stable and meaningful training signal than the traditional GAN loss function, and has been shown to produce better-quality images.

The combination of the U-Net and WGAN has been explored in recent years for various medical image analysis tasks. One common approach is to use the U-Net architecture as the generator in a WGAN framework for image synthesis, such as generating synthetic MRI images for data augmentation [32]. Another approach is to use the WGAN as a loss function in the U-Net architecture for improved segmentation accuracy, such as in the segmentation of the retinal layers from optical coherence tomography (OCT) images [33]. Overall, the combination of the U-Net and WGAN has shown promising results in various medical image analysis tasks, and further research in this area is ongoing.

Ronneberger et al. (2015) [12] introduced a convolutional neural network architecture called the U-Net for medical image segmentation. The U-Net architecture consists of an encoder, a decoder, and a skip connection between them. The encoder downsamples the input image and extracts its features, while the decoder upsamples the feature maps and generates a segmentation mask. The skip connection between the encoder and the decoder allows the model to retain high-resolution features, improving the segmentation accuracy. The U-Net architecture has been widely used in medical image analysis tasks, such as brain tumor segmentation [34] and retinal vessel segmentation [35]. Based on previous studies, we propose an improved model combining the U-Net and WGAN’s advantages.

## 3. Proposed Model

In this chapter, we introduce our U-Net-based WGAN model. The full WGAN source code used in the proposal model is available at https://github.com/choib/waegan_pl (accessed on 7 July 2024).

### 3.1. Advanced U-Net

#### 3.1.1. Entire Structure

Figure 1 shows the entire structure of the neural network modified based on the U-Net structure proposed in this study. The U-Net can be divided into encoders and decoders. The encoders and decoders consist of five stages each, and each stage maintains the original model [36] synthesized through the bypass. The modified U-Net we propose inserts an encoder part and a ResNet block, as shown in Figure 2. The modified part is as follows.

#### 3.1.2. Downsampling

We configured the encoder by connecting the convolutional neural network in parallel, without a nonlinear activity function such as the ResNet18 [37] and the ReLU. ResNet introduces various models such as the ResNet101 [38] and ResNet18, and the higher the neural network and the more parameters, the higher the classification accuracy. The parameter number of the entire network is a significant variable that increases the calculation, so the minimum size of ResNet18 is selected. The output image is reduced to one-half of the input image when passing through each block. However, the convolution for the d6 and d7 nodes in Figure 1 is performed without a size change, and only the number of channels is changed. The ResNet18 is provided with five blocks. The downsampling neural network corresponds to the output of each block. The input images are entered in the ResNet18 and the downsampling part. The output of the fifth block is branched into the U-Net’s upsampling and the encoder output unit. The encoder output (l7 and l8) is converted and used as a one-dimensional tensor through pooling. The output l7 for multiple class classification outputs nodes larger than the class numbers through the fully connected layer or dense layer. The encoder’s output supports single-node outputs (adversarial) and multi-class outputs (auxiliary). The learning method of each output is described separately.

#### 3.1.3. Upsampling

The encoder and the decoder, which looks like a mirror of the encoder, are configured. They consist of five blocks, and every time a block is passed, the images are doubled. The attention layer is introduced here. The attention layers or blocks used are limited to the image segmentation. The attention block improves the quality of the image segmentation results. This introduces the lower part of Figure 1. Each node in the downsampling part passes through the ResNet block and connects to the node of the upsampling part.

The ResNet block (or residual block) refers to a connector of the default unit. The basis of the U-Net structure uses the detour route (*F*(*x*) + *x*) used in this introductory unit. The nodes of the downsampling part and the nodes of the upsampling part are connected to this similar bypass. This unit structure is the basis of the ResNet structure, including ResNet18. A residual block is defined and used to connect the basic unit in plurality. The residual block has the disadvantage of increasing the training time due to the increased parameters as the number of connection units increases. However, the image segmentation quality tends to improve as the number of connection units increases.

#### 3.1.4. Auto Encoder and Dimension Reduction

The l7 of Figure 1 is output as an encoding result of the U-Net encoder as a latent vector. A latent vector with 32 dimensions is assumed as a default. Each dimension is expected to represent the features of the input image. However, it is difficult to grasp what each dimension represents as a physical feature.

The output value of the autoencoder is a latent vector. It is expressed as a combination of scalar values corresponding to the number of dimensions. To take advantage of this, the dimension reduction technique [39] is used. In this study, a fully connected layer is used to be expressed as one scalar value. As with the above, the activity function is not used in the autoencoder and dimension reduction. Softmax or Sigmoid is applied when calculating the loss function or using the scalar value.

A GPU is required to train the proposed neural network, and the distributed data-parallel (DDP) [40] method must be used to improve the learning speed. To do this, we used Pytorch-Lightning [41,42]. The neural network learning method suggested in this study is used in two ways. The WGAN is used as a learning method for pixel-unit image segmentation, and CNN-based discrimination is used for this. The multiple class classification uses the U-Net’s downsample part. Therefore, since the U-Net contains an inner discriminator, there is the characteristic that the generator and the discriminator learn at the same time during the neural network learning process.

### 3.2. WGAN

#### 3.2.1. WGAN

The WGAN is known as a learning method that contrasts with the learning method known as vanilla GAN [27]. Unlike vanilla GAN, which mainly calculates probability, it is characterized to deal with continuous values for the target. Assuming the value of the loss function changes slowly, it is a method of adjusting the amount of updates of the weight of the neural network with the clipping technique. Gradient exploding and mode collapse, known as problems with vanilla GAN, are less likely to occur. When learning the discrimination, the weight clipping is executed. The generator does not do weight clipping. It can be clipped off of the gradient of the entire model, but we do not use it in this study.

#### 3.2.2. Autoencoder

Studies have been conducted on multilabel semantic segmentation using the U-Net [32,43]. The autoencoder was proposed to implement active learning. It branches the nodes at the end of the downsample part of the U-Net, includes a dummy image in the training dataset, and then assigns and learns a different class from this learning dataset. The U-Net’s encoder learns to distinguish the classes, including classes without data in the ground truth. The result of this learning is used as an indicator to predict the accuracy of recognizing a particular object. The Adversarial output, which distinguishes the synthetic image from the original image, is learned by the binary cross entropy. However, when using it for prediction, it can read an output with a continuous value and quantitatively express the degree close to a specific object.

A noisy image for encoder learning is an image that distorts the original image. The distorted function used at this time uses the random affine function of PyTorch’s torchvision package. The encoder learns to be similar to the distorted image with affine and the output value for the input image. In the image distorted by affine, it was attempted to learn to recognize particular objects accurately, similar to the input images. The autoencoder learns the latent vectors are the same. The final output of the generator is a segmented image, to learn the original image ground truth and MSE to minimize. On the other hand, the structural similarity of the segmented image and the original image is considered, and this ratio is designed to be changed in proportion to the number of learning. An encoder loss is calculated during the discrimination learning. Noisy input during the discrimination learning is used to synthesize the Gaussian random noise with the input image. The random noise is used to create an input without information. The latent vector extracted from the input image is information. The information when there is an object to be segmented and the information in the dummy image case are different (labels). The latent vector of the synthesized image with the Gaussian random noise should not be information. Therefore, the loss function is calculated compared to the random integer. The latent vector calculates the loss function by comparing the result of the input image’s dimensional reduction to the valid (1.0) value and by comparing the random noise to the fake (0.0) value, both of which are used in calculating the loss function.

#### 3.2.3. Loss Function

The training process of the neural network is to update the parameters associated with the weight of the neural network in the direction of reducing the result of the loss function. Loss functions can mainly be divided into functions that process discontinuity values, such as distinguishing categories, and functions that process continuous values. The functions that process continuous values are often used to calculate and use an average squared error (ASE) or mean absolute error (MAE). The loss functions that use discontinuity values accompany the probability calculations that will be included in the category or class. The Sigmoid or Softmax functions containing index functions may be used in this case. Even if the gradient of these values is calculated, it becomes an exponential function, and the speed of increase or decrease is fast. In the case of complex neural networks, there are cases where the total loss function value diverges. To avoid this risk, a new loss function was defined and used to harmonize with the WGAN learning technique.

Label smoothing [44]: Cross entropy as a loss function is designed to alleviate the rapid changes caused by the discontinuous functions. In addition, the MSE that calculates the distance instead of the cross entropy, which calculates the probability, is used as a loss function.SSIM [45]: It is one of the methods used to quantify the degree of structural similarity between images. The output images are used along with the MSE as a loss function to match the ground truth structurally.*nn.BCBWithLogitsLoss():* As a loss function provided by PyTorch, it combines the binary cross entropy (BCE) with the Sigmoid function. There is no need to apply the Sigmoid to the target variable. By using this loss function, the target variable can be treated as a scalar value after training. This scalar value, called validity, can be interpreted as a metric of the latent vector.

#### 3.2.4. Generator and Discriminator

In GAN training, the discriminator and the generator are often separated. If the learning of the generator and the discriminator can be separated, the number of learning can be different in consideration of the complexity and learning speed of each neural network [41]. In the case of the proposed neural network that uses some of the U-Net for discrimination, it is difficult to control the number of learning times of the discriminator and the generator, so it is possible to change the weights of the discriminator and the generator in an indirect manner. When setting the optimizer, we create the generation learning and discrimination learning separately, and we can adjust the number of executions by indicating ‘frequency’. We can also adjust the time of command using environment variables. The default is one. In this default setting, the loss functions considered during generation learning and the loss functions considered during discrimination learning are arranged to control the overall balance.

## 4. Experiments

The performance of the U-Net-based semantic segmentation model proposed in this study was proved by evaluating the provided labeling images (5171 + 3000 images). A total of 5171 data were provided in September 2021, and 3000 video images were received in November 2021.

### 4.1. Datasets

The dataset received from the host agency in September was separated by 4136:1035 in training and testing, and a total of 5171 labeled images were used for the pilot studies. The labeling was defined as the four larynxual structures of R-TVC, L-TVC, R-FVC, and L-FVC, as well as the three lesion classes of Subglottis, Benign Tumor, and Cancer. The dataset delivered from the host organization on 26 November 2021 was separated into 2700:300 in training and evaluation and used as additional data in this study. In this study, we used the datasets labeled with polygons with seven colors. The ground truth data were labeled by using the H value in the HSV color space based on the color table with the seven colors, as shown in Table 1.

### 4.2. System Setup

PyTorch 1.7 or higher, Pytorch-Lightning 1.4.9 or higher, and hardware that operates two or more GPUs must be installed to operate all the features of the open package. In the case of a single GPU, DDP and multi-GPU functions cannot be utilized. NVIDIA GPU GeForce 3090, DGX v1 Server, and DGX A100 Server are confirmed, operating in an OS with a driver and a library installed on AMP if we use AMP [46]. It is recommended to use the Ubuntu Linux OS; in the Windows 11 WSL2 environment, the AMP backend can be set to native, but it is not recommended because of the limitations of the AMP function.

### 4.3. Evaluation Methods

The indicators that evaluate the learning results depend on the purpose for using the model. Storing the weights and biases at every learning time exhausts the hard disks, so the parameters should be stored only in the number of learning times where the evaluation indicators improve. In this study, the MSE loss function is used as a main performance indicator because the purpose is to produce results close to the image of the original data with a trained neural network model. PyTorch-Lightning has the ability to record the loss functions used in the training process. It is also linked to Tensorboard 2.7 [47] to observe the recorded data in the learning process.

### 4.4. Observation

We used the Tensorboard with PyTorch-Lightning to specify a scalar value we wanted to observe, and usually opened a specific port with a web browser to observe in real-time. Figure 3 is an example of the Tensorboard output. The MSE loss is a good indicator that can display the learning results by comparing the results of the neural network prediction with the original data. Every time the learning ends, this indicator is compared to store the learning results when the numbers are smaller than before. The results of the learning are overwritten by the name *last.ckpt*, which extracts three good results and stores them separately. The scalar value recorded under the name “Validity” is a value that expresses the latent vectors as scalars. The validity value is displayed in three different colors; the color varies for each version recorded, and the version is different every time the Tensorboard is called. If the training is discontinued and resumed, the version is different. The figure above is the result of the learning with different datasets. It shows that the mean value of validity varies from dataset to dataset.

### 4.5. Experimental Results

The model’s performance is tested using a dataset different from the one used in training. The test dataset is initially prepared and separated from the training dataset. After the batch work for the evaluation, the confusion matrix [40] is calculated to evaluate the model’s performance. Since the output of the neural network model is an image, and the class is assigned according to the color so that the data for the ground truth is created, this model is designed so that the confusion matrix is calculated separately according to the color. After the model training is completed, the process of performing predictions for evaluation and comparing the result images with the original data is preceded as an evaluation process. After the evaluation is completed with the batch work, the confusion matrix is calculated separately. The formula for calculating the confusion matrix is the same as the formula applied to known packages such as scikit-learn [48], reflecting the variables specific to this study, such as image size. The confusion matrix calculation combines and calculates several functions of the OpenCV [49] package, which deals with the image on four number values such as TP, TN, FP, and FN. At this time, the color expression of the three-channel RGB format used in the image is converted to the HSV format inside the code. Based on the extracted numbers, the necessary values, such as the F1 score and accuracy, are calculated and output. Meanwhile, for the intersection over union (IoU) [50], the bbIoU calculates the area with a traditional box and the IoU calculates the area with a polygon (pixels and boxes) and the output at the same time.

In addition to the numbers (F1, TP, TN, FP, FN, ACC) obtained by the confusion matrix calculation, the IoU, bbIoU, and approximation (mAUC) of the AUC [51] value that can be extracted from each image are also output. In addition, the difference between the encoder Gram Matrix value of the images distorted in the input image and the specific conditions is generated. Validity is a scalar value output that executes the dimensional reduction on the latent vectors. It is printed with standard deviation and a ratio of the same area as the target value (HSV) in the HSV image created by the neural network. The number output is the average value tested using the entire test dataset, and the standard deviation is output side by side with the average value and validity. In addition, the value is arranged in a large order among the nodes (32 default values) of the latent vector, indicating the first, second, and third indexes and counts. In the example below, there are 86 out of 206 samples, meaning that Node 28 of the latent vectors has the largest value, and Node 25 has the second largest value, with 49 samples. It is inserted to study the role each latent vector plays in the image creation.

Below is a list of the files and directories generated during the prediction. The name of the file and directory is created by the environment variable date and dataset. For example, Figure 4 shows the predicted class, the top three latent vector indexes, the difference value in the latent vectors, the validity of the dimension-reduced latent vectors, the encoder error, the number of colors, the detected polygonal area size, the polygon classified into the class and the area ratio of all the detected polygons, the central coordinates of the detected polygon, and the position of the input image are displayed. In addition, this model stores the polygon’s coordinates of the maximum size of the detected polygon. The JPG directory stores an image that combines the output and input images of the model. In the PNG directory, an image of a polygon with the maximum detected size is saved. Figure 5 shows that the images stored to check the progress after each epoch are gathered. From the left, the original data, the segmented result, and the input data are synthesized, and the input data are shown in the batch size. The data, 373, used at this time are the evaluation data that are not used for training. The PyTorch-Lightning modules, 374, store the smallest epoch in the MSE loss as the checkpoint file after the learning, 375, is terminated. These files are read and used for the test. There are many indicators that 376 shows the results of the evaluation, which consider that the F1 and IoU values in the 377 dataset are the most suitable. In Table 2 below, the class indicators of September (1035 tests, 378 samples) and November (300 test samples) are shown. In the case of the IoUs, the pixels, 379, are counted and calculated (IoU pix), the boxes are calculated, and all of them are shown. The peculiarity, 380, is that the indicators of Benign Tumors (Class 0) are lower than cancer’s 381 indicators. In other words, the learned neural network detects benign tumors smaller than 382, the labeled original, and the high indicators of cancer can be determined to have good cancer detection performance, 383.

On the other hand, the indicators extracted from the confusion matrix are filtered at 20 intervals (corresponding to 40 degrees, maximum 360 degrees) in H values in the HSV color space, and the extracted image is counted in pixel units. The confusion matrix results of each class of 300 evaluation datasets (received in November) are as follows.

If we calculate the accuracy using the data from Class 6 in Table 3, it is as follows.
(1)$$Accuracy=\frac{2417.41+94,777.63}{2417.41+94,777.63+432.32+676.65}\;\;\;$$

The accuracy calculated from the Formula (1) is approximately 0.989 or 98.9%. Table 4 shows the calculation results of the confusion matrix for each class based on the evaluation dataset of 1035 samples received in September.

**Table 3 cancers-16-03482-t003:** The confusion matrix and IoU values were extracted from 300 data. This table is the evaluation results for the remaining 300 images after training 2700 video images among 3000 video images.

Cls	Count	F1	TP	TN	FP	FN	ACC	IoU	bbIoU	mAUC	Precision	Recall
6	300	0.77	2417.41	94,777.63	432.32	676.65	0.99	0.65	0.38	0.39	0.85	0.78
5	300	0.74	2670.96	94,389.35	479.80	763.89	0.99	0.62	0.46	0.37	0.85	0.78
4	300	0.72	871.88	97,029.50	166.04	236.59	1.00	0.55	0.42	0.35	0.84	0.79
3	300	0.74	977.20	96,889.34	215.41	222.06	1.00	0.59	0.39	0.39	0.82	0.81
2	300	0.64	409.78	97,535.78	140.10	218.34	1.00	0.33	0.28	0.33	0.75	0.65
1	300	0.65	396.02	97,564.11	123.58	220.30	1.00	0.35	0.29	0.32	0.76	0.64
0	300	0.54	125.57	97,775.43	68.55	334.45	1.00	0.14	0.19	0.20	0.65	0.27

## 5. Discussion

This study aimed to automatically classify laryngeal lesions from video images. To achieve this, advanced segmentation techniques were needed to accurately distinguish between the lesions and their surroundings. We proposed an improved U-Net architecture for pixel-based segmentation and enhanced the segmentation results using a WGAN. This proposed model demonstrated high accuracy in cancer detection, achieving an overall accuracy rate of 99%.

Upon closer examination, the ACC value of the larynx endoscopic video images showed an accuracy of over 98% in each table. Specifically, the cancer diagnosis experimental results using the dataset for Class 6 showed outcomes of 99% and 98%, respectively, demonstrating the effectiveness of the proposed segmentation algorithm for malignant tumor diagnosis. A low F1 score indicates a significant difference between the ground truth and the polygonal areas segmented by the neural network. However, it does not imply that the cancers were not detected. This difference suggests that the definition of the boundaries of an area can vary each time a human labels it. It is an error that may occur during human labeling, whereas the boundaries determined by the neural network through statistical training may have fewer repetitive errors. Therefore, the semantic segmentation of the laryngeal video can be considered successful, and it can be concluded that the proposed model is suitable for processing this data. When evaluating the F1 and IoU scores by class, the detection accuracy of benign tumors was relatively low compared to other anatomical structures and lesions. At the same time, cancer showed relatively high detection results. If the human labels are inaccurate in generating the ground truth, it may lead to issues with the training accuracy.

In order to further improve the performance of our model in the automatic classification of laryngeal cancer using a WGAN, several key directions should be explored in future studies. First, expanding the imaging modalities used in the training process is essential. Our dataset consisted of frontal images of the larynx, which provided only a limited view of the lesions. However, laryngeal cancer can manifest in complex three-dimensional patterns that are not fully captured by a single viewpoint. To address this, future research should include images taken from multiple angles using endoscopic techniques, allowing the model to understand the spatial characteristics of the lesions better. Additionally, incorporating three-dimensional imaging data, such as reconstructions based on CT or MRI, would provide crucial depth information, allowing the model to assess both the surface and the depth of cancerous tissue. Advanced imaging techniques like narrow-band imaging or fluorescence endoscopy should also be considered, as they enhance the visualization of abnormal blood vessels and tissue, further improving the model’s ability to distinguish between healthy and cancerous regions.

Another critical aspect of future work involves automating the lesion segmentation process. Currently, the model relies on manually annotated data, which, while highly accurate, is time-consuming and limits scalability. To overcome this challenge, future studies could incorporate existing deep learning models, such as the U-Net or Mask R-CNN, to automatically segment the lesions in laryngeal images. Once trained on a large and diverse dataset, these segmentation models could significantly reduce the need for manual annotations. Moreover, self-supervised learning techniques could be utilized to pre-train the models on unlabeled data, allowing them to learn valuable features before being fine-tuned for the segmentation tasks. This would enable the models to better use the vast amounts of unlabeled medical images that are often available in clinical environments. Weakly supervised learning is another promising avenue, as it allows models to be trained using only image-level labels, such as “cancerous” or “non-cancerous”, without requiring a detailed pixel-level segmentation. Techniques like Class Activation Mapping (CAM) or Grad-CAM could be used to highlight the areas of interest in the images, effectively automating the process of identifying the cancerous regions.

Beyond data acquisition and segmentation, improvements to the deep learning model architecture could yield further enhancements in the classification performance. Attention mechanisms, which enable the models to focus on the most relevant parts of an image, could be integrated into the WGAN framework to detect cancerous lesions more precisely. Additionally, multi-task learning, where the model is trained to perform multiple related tasks simultaneously, could be explored. For example, in addition to classifying whether an image contains cancerous tissue, the model could predict the cancer stage or estimate the lesion’s size and depth. This approach would encourage the model to learn more comprehensive and robust features, improving its overall performance. Moreover, generative data augmentation, where the WGAN architecture is used to create synthetic images of laryngeal cancer, could help address the challenge of limited real-world data. By generating diverse and realistic examples, the model’s generalization ability could be enhanced, leading to a better performance in clinical settings.

Finally, it is critical to integrate the developed model into clinical practice. This would involve conducting large-scale clinical trials to validate the model’s performance across diverse patient populations and in real-world environments. Additionally, developing a user-friendly interface that allows clinicians to upload images and receive automated classifications quickly would facilitate the adoption of the model in medical settings. By incorporating these advancements, the automatic classification of laryngeal cancer using deep learning could become a powerful tool in early diagnosis and treatment planning, ultimately improving patient outcomes.

## 6. Conclusions

Semantic segmentation is a computer vision task that classifies each pixel in an image according to predefined classes. This study extends semantic segmentation to open-set areas and proposes a U-Net-based WGAN model. Through this model, we demonstrate a stable performance of 99% on the laryngeal cancer dataset. This technology has the potential to evolve beyond simple object segmentation, enabling the identification of various objects and regions to provide a more comprehensive description of the image. At this point, we believe it can help reduce the misdiagnosis rate caused by human error and enrich human lives through early diagnosis. Lastly, we think this research’s findings can be easily applied to other fields beyond laryngeal cancer diagnosis.

## Figures and Tables

**Figure 1 cancers-16-03482-f001:**
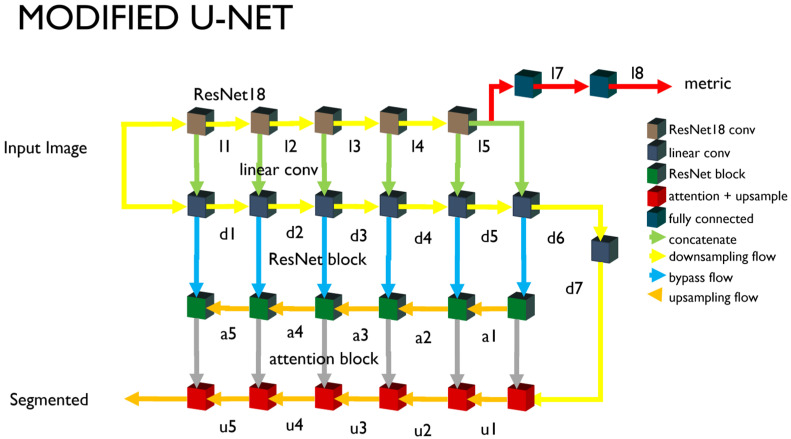
Schematic diagram of modified U-Net structure. ResNet18 blocks are inserted for latent vectors.

**Figure 2 cancers-16-03482-f002:**
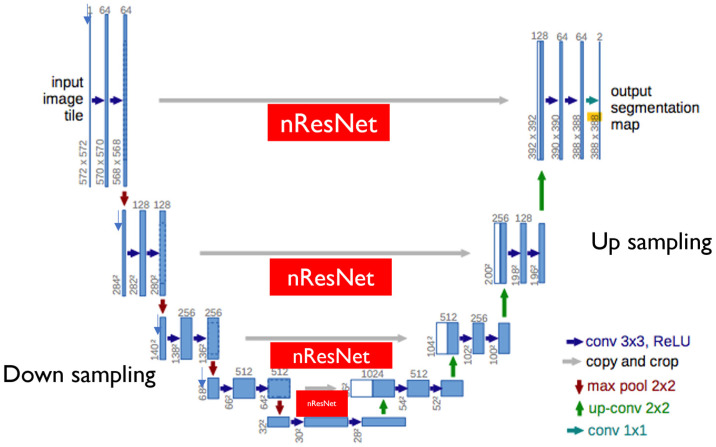
Our modified U-Net encoder structure. Besides skip connections between corresponding encoder and decoder modules, ResNet18 is included in our modified U-Net.

**Figure 3 cancers-16-03482-f003:**
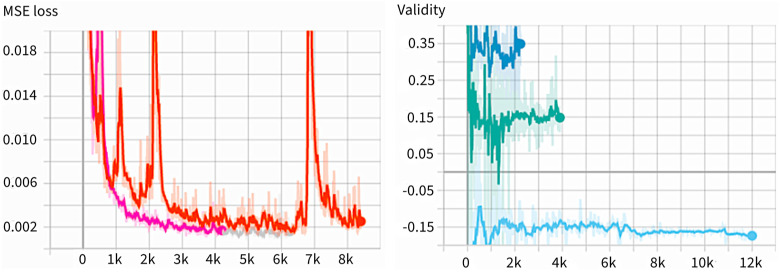
Example of Tensorboard output in MSE Loss (**left**) and validity (**right**). In validity, different colors indicate different versions of validity, showing training is discontinued and resumed.

**Figure 4 cancers-16-03482-f004:**
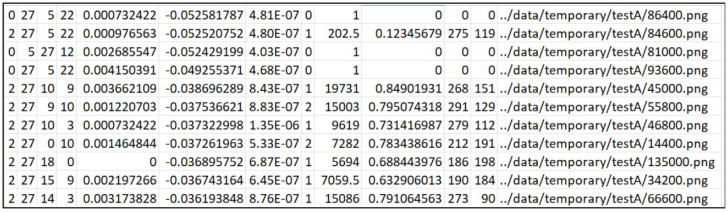
An example of a CSV file written during prediction. Each column represents the following: the predicted class (C1), the top three latent vector indexes (C2–C4), the difference value in the latent vectors (C5), the validity of the dimension-reduced latent vectors (C6), the encoder error (C7), the number of colors (C8), the detected polygonal area size (C9), the area ratio of the classified polygon and all the detected polygons (C10), the *x* and *y* coordinates of the detected polygon’s center (C11–C12), and the path of the input image (C13).

**Figure 5 cancers-16-03482-f005:**
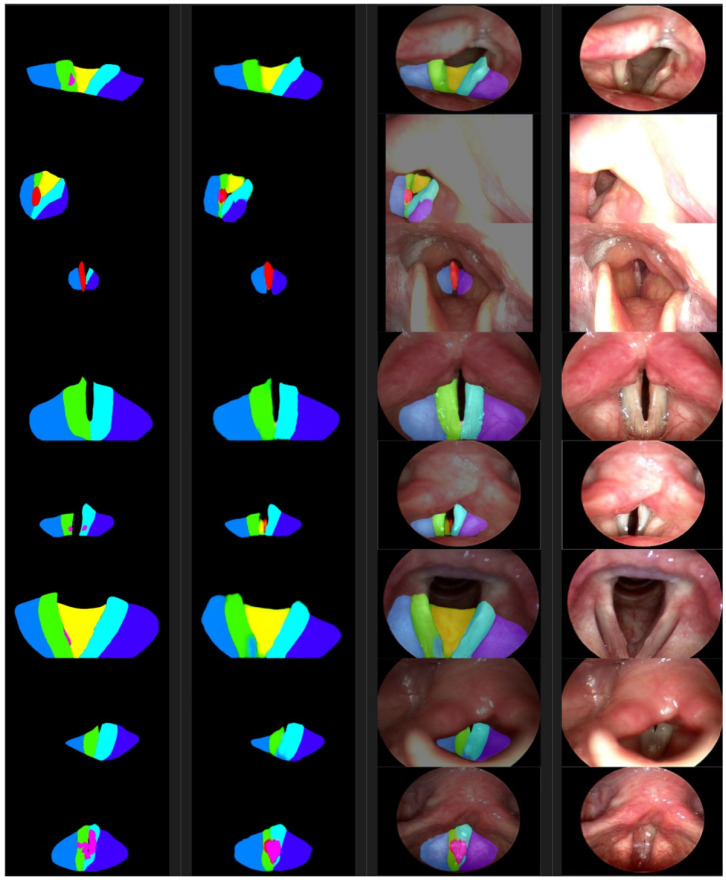
From the left are the ground truth examples, segmentations, photosyntheses, and photographic images. The color of a segmented area represents the class as defined in Table 1.

**Table 1 cancers-16-03482-t001:** Lesion and color table containing hexadecimal color codes.

Lesion	(H, S, V)	(R, G, B)		Color Code
R-TVC	(105°, 100%, 100%)	(64, 255, 0)	polygon color	#40FF00
L-TVC	(180°, 100%, 100%)	(0, 255, 255)	polygon color	#00FFFF
R-FVC	(210°, 100%, 100%)	(0, 128, 255)	polygon color	#0080FF
L-FVC	(255°, 100%, 100%)	(64, 0, 255)	polygon color	#4000FF
Subglottis	(60°, 100%, 100%)	(255, 255, 0)	polygon color	#FFFF00
Benign Tumor	(0°, 100%, 100%)	(255, 0, 0)	polygon color	#FF0000
Cancer	(300°, 100%, 100%)	(255, 0, 255)	polygon color	#FF00FF

**Table 2 cancers-16-03482-t002:** Training results were evaluated with 300 + 1035 data.

	Counts	300	1035	300		1035	
		F1	F1	IoU(pix)	IoU(BB)	IoU(pix)	IoU(BB)
Name	Class						
R-TVC	2	0.64466	0.65400	0.33159	0.28494	0.41680	0.41015
L-TVC	3	0.74483	0.72868	0.59073	0.39312	0.56017	0.54482
R-FVC	4	0.72005	0.68070	0.55318	0.41790	0.51202	0.53456
L-FVC	5	0.73936	0.72070	0.61976	0.46260	0.59198	0.55988
Subglottis	1	0.65393	0.65011	0.34636	0.29183	0.44008	0.43077
Benign Tumor	0	0.54029	0.50486	0.14321	0.18924	0.15398	0.22279
Cancer	6	0.77319	0.75490	0.64823	0.37732	0.60632	0.48714

**Table 4 cancers-16-03482-t004:** The confusion matrix and IoU values were extracted from 1035 data. 4136 of the 5171 data were trained and tested with the remaining 1035 video images.

Cls	Count	F1	TP	TN	FP	FN	ACC	IoU	bbIoU	mAUC	Precision	Recall
6	1035	0.75	2814.15	93,923.13	697.20	869.52	0.98	0.61	0.49	0.38	0.80	0.76
5	1035	0.72	3140.11	93,426.73	831.94	905.22	0.98	0.59	0.56	0.37	0.79	0.78
4	1035	0.68	861.58	96,943.25	228.44	270.73	0.99	0.51	0.53	0.33	0.79	0.76
3	1035	0.73	935.46	96,909.83	235.43	223.27	1.00	0.56	0.54	0.36	0.80	0.81
2	1035	0.65	586.78	97,251.42	245.84	219.95	1.00	0.42	0.41	0.33	0.70	0.73
1	1035	0.65	578.17	97,275.17	229.28	221.37	1.00	0.44	0.43	0.32	0.72	0.72
0	1035	0.50	128.96	97,857.93	70.05	247.06	1.00	0.15	0.22	0.17	0.65	0.34

## Data Availability

The original data used in this study were supported by Korea University Hospital and MTEG at https://medicine.korea.ac.kr/en/index.do (accessed on 26 November 2021).

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
