# Peer review of "Enhanced WGAN Model for Diagnosing Laryngeal Carcinoma"

_cancers, 2024, doi:10.3390/cancers16203482_

Round 1
Reviewer 1 Report
Comments and Suggestions for Authors
Introduction
First Paragraph has no citations. The first two sentences may flow better if combined. The first few paragraphs seem too broad for such a specific question trying to be answered in this manuscript.
Paragraph two can likely just be removed, it does not add too much to the manuscript. Paragraph Three: there are multiple areas where citations should be utilized.
"Here, among neural network models known 42 to be suitable for image classification, U-Net is used as a pixel-level image segmentation 43 device and reports the results[2,3]." can likely omit reports the results.
"Laryngeal cancer is a serious and potentially life-threatening disease that affects the 45 larynx, also known as the voice box. Early diagnosis and treatment of this cancer are 46 crucial for improving patient outcomes and survival rates. Modern technology has played 47 a significant role in advancing the diagnosis and treatment of laryngeal cancer. In this 48 paper, we will explore some of the latest technological advancements in laryngeal cancer 49 diagnosis and introduce related research papers." There are no citations in this paragraph. Laryngeal cancer should be brought up earlier in the discussion and spent more time on since that is the point of this manuscript.
Related Works
"AI has shown great promise in improving the accuracy and efficiency of a laryngeal cancer diagnosis." - needs a citation
Section 2.1 - the word "developed" is used too often.
"AI technology has shown great promise in improving the accuracy and efficiency of 82 a laryngeal cancer diagnosis." this sentence in some shape or form has been repeated multiple times so far. Considering being more concise.
Discussion: should start with a sentence explaining the point of this manuscript and the overall outcomes. Same with conclusions.
Comments on the Quality of English Language
There needs to be attempts to organize each paragraph and cut down on some of the repeated sentences. The introduction should follow a funnel format and the discussion should emphasize the importance and question that the project is trying to answer.
Reviewer 2 Report
Comments and Suggestions for Authors
This paper presents an enhanced WGAN method to diagnose laryngeal cancer. The method is sound. However, there are some concerns about this study.
- Please enhance the motivation/background on clinical aspects in this study
- Please well summarize the main contributions in this study.
- More explanation on the figures, implementation details are needed.
- Please further enrich the limitation and future work in this study.
- There are some problems on paper writing.
Comments on the Quality of English Languageminor
Round 2
Reviewer 2 Report
Comments and Suggestions for Authors
No further question.
Comments on the Quality of English LanguageMinor revision